# Comparison of Urban versus Industry Normative Values of Immediate Post-Concussion Assessment and Cognitive Testing (ImPACT™)

**DOI:** 10.3390/ijerph21030247

**Published:** 2024-02-21

**Authors:** Tamerah N. Hunt, Megan Byrd

**Affiliations:** 1School of Health and Rehabilitation Sciences, The Ohio State University, Columbus, OH 43210, USA; 2Department of Health Sciences Kinesiology, Georgia Southern University, Statesboro, GA 30458, USA; mmbyrd@georgiasouthern.edu

**Keywords:** concussion, social determinants of health, adolescent, low SES, baseline testing

## Abstract

Concussion baseline testing has been advocated for the assessment of pre-morbid function. When individual baseline scores are unavailable, utilizing normative values is recommended. However, the validity of generalizing normative data across multiple socioeconomic environments is unknown. Objective: mimic the normative data creation of ImPACT™ to examine the effect of socioeconomic status (SES) on ImPACT™ composite scores. Methods: A retrospective cross-sectional design analyzed completed computerized neuropsychological test data (ImPACT™) obtained to establish the baseline scores of cognitive function from males aged 13–15 years (*n* = 300) and 16–18 years (*n* = 331) from an urban high school system. Comparisons between baseline scores and normative ImPACT™ values were calculated utilizing *t*-tests with ImPACT™ composite scores serving as dependent variables. Results: significant differences between age-dependent urban composite scores and ImPACT™ normative values for 13–15- and 16–18-year-olds were found for Composite Verbal Memory, Composite Visual Memory, Composite Motor and Composite Reaction Time (*p* < 0.01). Conclusions: Significant differences exist between urban high school athletes and ImPACT™-provided age-dependent normative scores, with urban participants performing below age-dependent normative values. These findings support establishing SES appropriate normative values when baseline test scores are not available for direct comparison in order to provide better evaluation and post-concussion management across diverse populations.

## 1. Introduction

Concussions have reached near-epidemic proportions in contact sports at both professional and amateur levels; there are an estimated 1.6 to 3.8 million sport-related concussions occurring in the United States annually [1]. There has been a substantial rise in concussions among the adolescent age group highlighting a need for more age specific research [2]. Previously, clinicians believed that adolescent athletes mimic their collegiate counterparts in recovery patterns and neuropsychological deficits (i.e., brain fog, sleep disturbances) following a concussion. However, evidence suggests that the maturing adolescent brain may need to be treated uniquely [3,4,5,6,7,8,9]. Information processing, attention, motor and memory scores for adolescent athletes are lower than their collegiate and professional counterparts [7,9]. Further, studies have found that high school athletes on average take longer to recover (4–10 days) and report an increase in symptoms when compared to collegiate athletes following a concussion [7,10,11,12,13]. These findings support the potential need to change concussion assessment paradigms for youth athletics.

With the heightened awareness of concussive injuries in sports, many professional medical organizations have issued position statements for the assessment and management of concussions [14,15,16,17]. Many of these position statements recommend the use of baseline testing [14,15]. Estimates of pre-morbid ability (i.e., pre-injury status) are essential in making neuropsychological inferences. Such estimates are commonly obtained from nationally standardized measures of intellectual ability on high school transcripts, as well as aptitude test scores [18]. Understanding the quality of pre-morbid performance also provides an important context to view present neuropsychological performance.

The most common approach to the concussion-related assessment model has been to use a brief neurocognitive test battery, which typically measures memory, cognitive processing speed, working memory, and/or executive function prior to and following a concussive injury [19,20]. One of the strengths of the sports concussion assessment model is the emphasis on assessment of pre-injury assessment or baseline function [5,8]. This model has been advocated extensively to obtain a pre-injury baseline assessment of all athletes at risk of incurring a concussion during sports participation. These baseline scores are important so that a comparison can be made between pre-injury and recovery from a concussive injury.

When baseline measurements are not available, the use of age-appropriate normative data is suggested [21]. The primary limitation of using normative data is overlooking the subtle changes related to individual differences during concussion assessment when comparing to age-dependent industry-provided normative data. Second, normative data are only valid in the population for which it is obtained [21,22]. If the provided normative values are based upon a population that is not representative of the individual being assessed, accurate interpretations in test performance following a concussive injury are difficult to impossible.

Many factors may influence test performance such as distractions during testing, failure to understand directions, fatigue and socioeconomic status [23]. There is increasing evidence showing that pre-injury conditions, family environment and the level of development prior to injury can predispose children to neuropsychological and postural deficits [3,24,25,26,27,28]. While factors such as learning disabilities, previous history of concussion and effort have been examined and identified as gross factors that are correlated with poor performance on cognitive and balance assessment [3,11,29], socioeconomic status has been neglected in sports-related concussion research.

Socioeconomic status (SES) is an economic and sociological combined term that is characterized by many encompassing factors that may include income level, educational level, and occupation [30]. In 2005, the World Health Organization provided a framework that posits that health inequality is a product of social stratification, which in turn, leads to certain groups being put at further risk of disease or injury [31]. Evidence suggests that cultural beliefs may be a significant factor for poorer outcomes following injury in this population [32,33].

Individuals that are considered to have lower SES have greater incidence of heart disease, lung cancer and diabetes, with explanations due to a poor diet, decreased exercise and higher levels of smoking [32,33,34]. SES may also impact parental use of family resources to enrich developmental experiences with hobbies, recreation, museums, libraries, and travel, etc. [33]. Moreover, it appears that SES may affect other dimensions of parenting such as emotional and verbal responsiveness of the parents as low-SES families may have limited resources to offer reinforcement for desired behavior to encourage the development of executive skills [33].

Neurocognitive outcomes following injury have been predicted by less advantaged family environment [35,36]. Cognitive and academic abilities in children with severe traumatic brain injury (TBI) show limited recovery during the first-year post injury. Following the first year, there is a slowing of any continued catch-up growth [37,38]. Measures of environmental disadvantage predict lower scores on most tests in children with and without TBI [39]. Data from 131, 461 TBI patients showed that race and SES race were associated with differences in mortality, LOS, and discharge to inpatient rehabilitation [40].

Higher socioeconomic status (SES) is positively correlated with performance on most tests in youth populations [36] and so there is precedent that SES may influence test scores. Although measures of environmental disadvantage predict lower scores on most tests in children with and without TBI, these factors may amplify the effects of traumatic brain injury on tests. Differences exist between children living in households of low and high socioeconomic status, particularly on scales of language and executive function [33,34,41]. Therefore, the research consistently finds that children living in higher SES environments show an increase in cognitive performance, while those in lower SES maintain a flat performance.

While the theoretical framework that low-SES patients respond differently to injury, evaluation and recovery is not new, the causes and factors associated with this difference are vast and the effect of SES on concussion outcomes is poorly researched. Consistent, although scarce, research has identified differences between those of high and low SES status on neuropsychological test scores and recovery [42,43]. Measures of environmental disadvantage (i.e., limited access to resources in public spaces) predict lower scores on most tests and poorer quality of life in children with and without TBI [44]. Available research establishes consistent findings that SES effects neuropsychological test scores and recovery, with low-SES participants performing worse on cognitive test scores compared to their high SES counterparts [42,43]. Further, differences exist between low and high SES on time to recovery with high SES taking longer to return to sport and learn [42,43].

The effect of socioeconomic status during testing has been extensively studied in education and intelligence testing. SES, however, has been under-tested in sports-related concussion assessment and management. The increase in athletic participation demands that health care practitioners understand the implications and effects of the environment on neuropsychological tests utilized in concussion assessments. Investigating concussions from the perspective of socioeconomic status and using this framework could assist in developing relevant assessment tools and clinical management protocols. The aim of this study was to compare baseline computerized neuropsychological composite scores from urban high school athletes to the normative values published by ImPACT™. Further, percentile ranks were calculated for each ImPACT™ composite score to begin the establishment of normative values available for an urban low-SES school district. It is hypothesized that a low socioeconomic urban school district may have lower normative values and current program developed normative values may not be accurate for low-SES urban school populations.

## 2. Materials and Methods

Participants: A sample of junior varsity and varsity football players aged 13–18 from several high schools within a low socioeconomic urban school system were enrolled in the study (Table 1). The sports medicine clinic provides athletic training outreach services, inclusive of concussion assessment and management, to numerous school systems in the area. The school system has approximately 55 thousand students in 125 schools. This school system is defined as urban, low SES, and a Title I school due to the location and demographic characteristics of the population it serves. School districts are designated as Title I by a federal education program that supports low-income students throughout the nation. Funds are distributed to high-poverty schools, as determined by the number of students that qualify for free and reduced lunch. The school system is made up of a 28% Caucasian and 70% African American student population. A total of 81% of students receive free or reduced lunch and the average family income is $36,287 with a parental education of ninth grade. Exclusionary criteria included any self-reported diagnosed learning disability, previous history of concussion, a diagnosed psychiatric disability, a repeated year in school, treatment for migraines and/or invalid baseline scores as determined by ImPACT™. Consistent with ImPACT™ normative categories, participants were divided into two age categories, 13–15-year-olds and 16–18-year-olds [13,45]. The participant numbers in this study far exceeded the numbers for ImPACT™ normative data which were based on 183 boys aged 13–15 and 158 boys aged 13–18. [45] Subsequent normative data sets contain less than 150 participants per group [13,45].

Main outcome measures: ImPACT™ composite scores served as dependent variables. ImPACT™ is a commonly utilized computerized neuropsychological test within concussion assessments [13,45,46]. ImPACT™ is a brief internet-based neurocognitive assessment battery. ImPACT™ is composed of six tests that measure three speed indices: simple reaction time, complex reaction time and speed of information processing. Quantitatively, ImPACT™ yields four clinical composite scores, including verbal memory, visual memory, visual motor speed, and reaction time. ImPACT™ also contains the impulse control composite, a validity index score. ImPACT™ also has a brief questionnaire and symptom checklist to obtain additional demographic information. ImPACT™ has been shown to be a valid measure for assessing the neurocognitive effects of sports-related concussions.

Data collection procedures: Prior to the competitive season, all athletes are required to perform Pre-Participation Examinations (PPEs) prior to their competitive season which are valid for the entire one year. The state required PPE includes a thorough personal and family history form and physical examination which includes height, weight, blood pressure, vision assessment, and review of systems inclusive of ears, eyes, nose, throat, heart, lungs, abdomen, skin, and neurological and musculoskeletal systems. Mass PPEs were conducted by physicians, nurses and athletic trainers from the sports medicine clinic at the medical facility or school. Following PPEs, computerized neuropsychological testing was conducted at the respective high school computer lab to establish a baseline prior to competition. The computerized neuropsychological platform utilized was the Immediate Post-Assessment of Concussion Test (ImPACT™) which was administered as part of the standard of care for concussion assessment prior to the start of their competitive season. During the baseline assessment, the participants completed a health questionnaire to obtain demographic information, concussion history, preexisting neurological conditions and evidence of other medical conditions. Following the health questionnaire, participants completed the symptom checklist and six cognitive tests (described above) which is consistent with the previous literature [13,45]. After exclusionary criteria were ruled out, the remaining participants’ neuropsychological battery and self-reported symptom assessment measures were included in the analysis. This research was conducted by reviewing records of all athletes that have completed the concussion management tool ImPACT™. All test administrations occurred in controlled areas free from noise and distraction and were provided under the supervision of a test administrator. Participants were tested in small groups, and athletes were positioned away from others to allow them to maximize their concentration. This study was approved by the University Institutional Review Board.

Statistical analysis: A total of 884 participants were initially obtained from the data set. Neuropsychological test data that contained any exclusion criteria were removed from the analysis; 253 participants were eliminated from the analysis. The breakdown of participants removed for exclusionary reasons are as follows:Previous history of concussion (*n* = 53)Learning disability (*n* = 35)Diagnosed psychiatric disability (*n* = 28)Repeated a year in school (*n* = 24)Treatment for migraines (*n* = 16)Deemed invalid by ImPACT™ (*n* = 97)

Statistics were calculated on the remaining 631 participants. Percentile ranks were calculated for each ImPACT™ composite score to establish the frequency distribution that are less than the score. Comparisons of groups to published norms were calculated by utilizing one-sample *t*-tests. Significance was set a priori at 0.0083 after the Bonferroni correction for six dependent variables. Statistical analyses were performed using STATA 10.1 (Stata Corporation, College Station, TX, USA).

## 3. Results

A total of 631 high school male participants with 300 participants aged 13–15 and 331 participants aged 16–18 years old were involved in the study.

A significant difference existed between age-dependent urban composite scores and ImPACT™ normative values for 13–15-year-olds for Composite Verbal Memory, Composite Visual Memory, Composite Visual Motor (processing speed) (Figure 1) and Composite Reaction Time (*p* < 0.01) (Figure 2).

A significant difference existed between age-dependent urban composite scores and ImPACT normative values for 16–18-year-olds for Composite Verbal Memory, Composite Visual Motor (processing speed) (Figure 3) and Composite Reaction Time (*p* < 0.01) (Figure 2). Consequently, lower SES, urban participants performed worse than the provided normative data. Moreover, the differences in scores are larger in the 13–15-year group compared to the 16–18-year-old group. See Table 2 for detailed statistics.

Percentile ranks were calculated for the normative and lower SES, urban population for the composite scores (Verbal Memory, Visual Memory, processing speed and Reaction Time). Statistical differences were not calculated between percentile ranks. But, there is a trend that the lower SES performed better in the lower percentile ranks; however, at or above the 50th percentile rank, the normative data was higher (Figure 4, Figure 5, Figure 6 and Figure 7).

## 4. Discussion

Baseline ImPACT™ scores were significantly worse in our sample of lower socioeconomic urban high school athletes. Differences were noted in composite scores for Verbal Memory, Visual Memory, Visual Motor and Reaction Time across all percentile ranks. Many factors may contribute to the differences between published normative data and individuals from a low socioeconomic urban environment. The family environments of children from low SES backgrounds are often characterized by organizational chaos, lack of structure and routine, exposure to multiple stressors [47], and excess background noise and crowding [48,49]. Researchers have estimated that regardless of the other confounding variable, home environment has been indicated as a mediator [50].

Clear delineation of environmental influences has not been established. Examination of specific factors such as cultural values, economic opportunities and age within SES provides some limited evidence of SES on neuropsychological testing. First, more traditional African American cultural values/beliefs (e.g., religious beliefs, cultural distrust and family values) were associated with lower overall neuropsychological test performance [51]. Studies have shown that there are consistent differences between children living in households of low and high socioeconomic status, particularly on scales of language and executive function [52]. Additionally, a case–control study of African Americans aged 10–13 found that those who were in the group with higher socioeconomic status performed significantly better on tests of language, working memory and cognitive control than those in the lower SES group [53].

Economic opportunities have been examined to assess their influence on cognitive function. Baxendle found that lower socioeconomic status may be a proxy marker for the limited economic opportunities associated with compromised cognitive function [54]. Ip found that rats reared in an enriched environment following injury recovered faster and eventually surpassed their pre-morbid level, while those raised in a non-enriched environment took longer to recover and did not obtain increased function following injury [55]. SES may impact parental use of family resources to enrich developmental experiences with hobbies, recreation, museums, libraries, and travel, etc. Moreover, it appears that SES may affect other dimensions of parenting such as emotional and verbal responsiveness of the parents. SES families may have limited resources to offer reinforcement for desired behavior and provide scaffolding to encourage the development of executive skills [52].

We found that our low-SES population exhibited larger differences in scores in the 13–15-year-old group compared to the 16–18-year-old group. The growing literature has examined the interaction between cognitive maturity and neuropsychological testing; the interaction between socioeconomic status is less developed. There have been studies that demonstrate an interaction between age and SES, showing worsening cognitive scores in the older participants between the higher SES group and the lower SES group [55]. Children living in higher SES environments show an increase in cognitive performance from ages 3–6 while those in lower SES maintain a flat performance. In our study, the younger participants performed worse than the older participants when compared to the published normative data. Most studies that demonstrate widening gaps as a function of age investigate younger populations which may not be replicated in adolescent participants. The investigation of computerized neuropsychological test scores should be examined to determine if there is an interaction between age and cognitive function as mediated by SES.

The cross-sectional design of the current study does not allow us to address issues of causality directly; the results are suggestive of socioeconomic mediators that play a role in neuropsychological test scores and should be examined further during the assessment and management of brain injury. This study provides additional evidence that wealth and education may operate differentially and independently on different neurocognitive systems. While researchers have had difficulty quantifying the magnitude of the effect of socioeconomic status on neuropsychological scores, the presence of poorer scores begs that clinicians be aware of the needs of the population with whom they work. These findings appear to be the first attempt to understand the effect of socioeconomic status and sports-related concussion assessment and management. Many interpretations from this research are limited by our cross-sectional approach with the information provided and obtained. However, as the first examination of SES in concussion assessment, our findings beg for longitudinal prospective research in this environment to determine the best assessment and management protocols for specialized populations.

### 4.1. Clinical Implication

Recent legislation requires medical care for adolescent athletes sustaining concussions. Clinicians must have a thorough understanding, inclusive of limitations, of assessment tools available to clinicians working across multiple populations. While most agencies and these researchers recommend a multifaceted approach to concussion assessment and return-to-play decisions, many clinicians rely heavily on neuropsychological testing. Clinicians should be aware that there are several cognitive assessments available for concussion assessment that include both traditional pencil and paper neuropsychological test batteries and computerized test batteries. While this research is not meant to recommend one specific neuropsychological platform, it strongly suggests evaluating the available programs to determine which testing platform and paradigm may be the most appropriate for their population. Implementation of personalized health care demands that health care practitioners understand the effects of the environment. Our findings may necessitate the establishment of age and SES appropriate normative values when baseline test scores are not available for direct comparison.

### 4.2. Limitations of the Study

There are several limitations that warrant comments. First, this study was a retrospective examination of scores. The participants were taken from baseline data collected as a part of the standard of care for concussions within a low-SES, urban, Title 1 school district where athletic training services were provided through a sports medicine outreach program. While this study was not initially hypothesis driven, the sample sizes were similar to the number of participants utilized for normative populations provided by ImPACT™. As the findings are based upon one low-SES, urban, title I school district, the results may not be generalizable to other environmental settings that include low-SES populations. Future studies should include data from additional low-SES, urban, Title 1 high school districts to enable generalizability across the nation.

Second, while we eliminated participants with commonly assessed confounding variables such as learning disabilities, previous history of concussion, mental illness and invalid test scores, our retrospective cross-sectional approach did not account for cognitive maturity, fatigue, academic ability and/or motivation. Evaluation of and attempts to eliminate the interactions among these variables within the high school population are necessary for future studies.

Finally, the sample was limited to male high school athletes that participated in football. There is evidence that gender may affect cognitive test scores with females performing better than their male counterparts [56]. Therefore, interpretation of these data should only be considered for male athletes. To enhance interpretations, additional studies should be completed that involve both genders across multiple sports.

## 5. Conclusions

The increase of media and clinician awareness of concussion assessment has increased the utilization of tools inclusive of computerized neuropsychological testing during baseline assessment. Significant differences existed between urban high school athletes and ImPACT™-provided age-dependent normative scores with urban participants performing below age-dependent normative values provided by ImPACT™. Historically, research involving SES has been prominent in intelligence and traditional neuropsychological literature. The effect of SES in mild traumatic brain injury is less understood.

The influence of race and ethnicity on standardized testing consistently lowers scores for minorities; however, the multidimensional and intersectional nature of SES (inclusive of race/ethnicity) makes it harder to directly examine causal relationships. This study provides initial evidence that if clinicians are working with a low-SES urban population and utilizing normative data for return to play, the low-SES urban participant may not achieve ImPACT-provided normative test scores even though they have fully recovered, creating an unnecessary delay in return to learn/play/work. Bridging the gap between research and clinical practice is the key to better management of sport-related concussions and improving return-to-play decisions. Ultimately, clinicians need to develop new concussion management strategies inclusive of a multifaceted approach to concussion assessment that accounts for influencing factors such as environmental factors, specifically socioeconomic status. The first priority should be to establish SES-appropriate normative values when baseline test scores are not available for direct comparison in order to provide better evaluation and post-concussion management across diverse populations. Second, clinicians need to demand that test developers incorporate exhaustive demographically (age, gender, race-ethnicity, SES, culture, and education) corrected norms. Finally, clinicians and researchers should prospectively examine SES and the impact on the athlete’s performance on the field and/or their recovery trajectories. Future studies should incorporate (1) a prospective approach to identify individual level SES via personal factors, and (2) a longitudinal study design to enable tracking of SES over time to establish causation more definitively.

## Figures and Tables

**Figure 1 ijerph-21-00247-f001:**
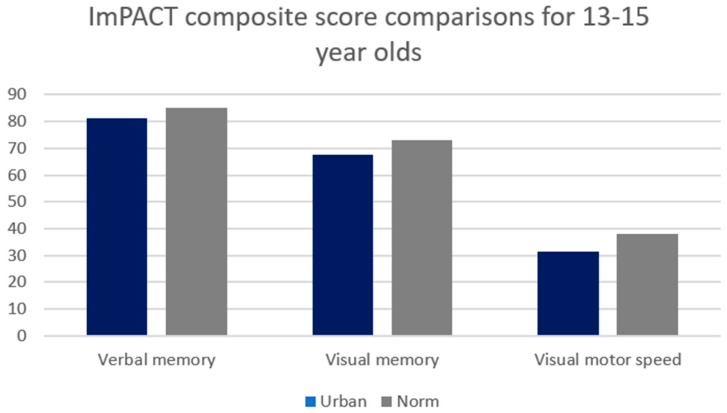
Comparison between 13–15-year-old urban high school group and industry-provided normative composite scores for Composite Verbal Memory, Composite Visual Memory and Composite Visual Motor. Significant differences existed between groups at the 0.0083 alpha level.

**Figure 2 ijerph-21-00247-f002:**
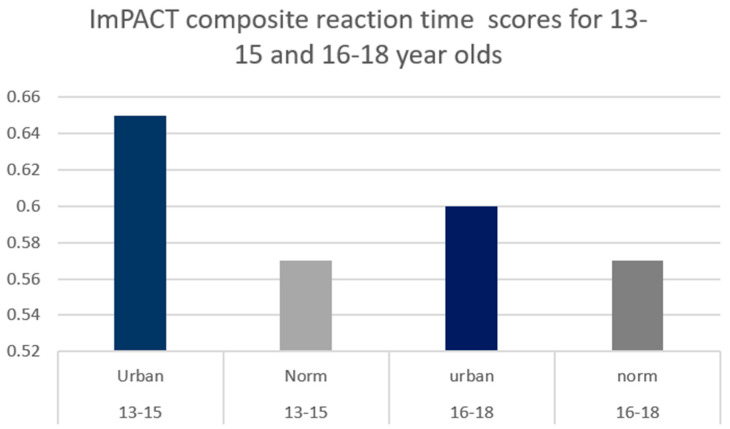
Comparison between urban high school group and industry-provided normative Composite Reaction Time scores. Significant differences were found between both age groups at the 0.0083 alpha level.

**Figure 3 ijerph-21-00247-f003:**
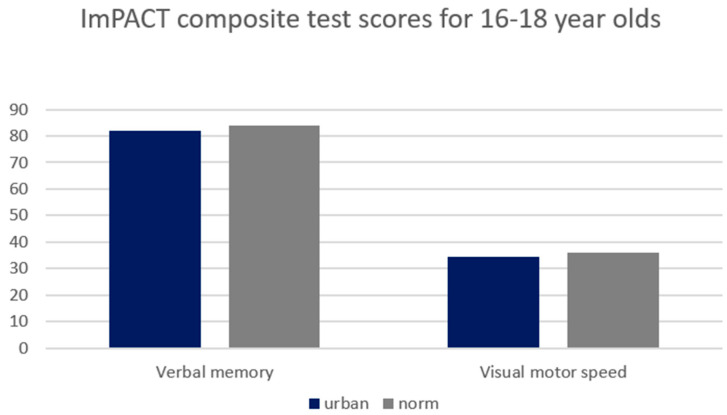
Comparison between 16–18-year-old urban high school group and industry-provided normative composite scores for Composite Verbal Memory, Composite Visual Memory and Composite Visual Motor. Significant differences existed between groups at the 0.0083 alpha level.

**Figure 4 ijerph-21-00247-f004:**
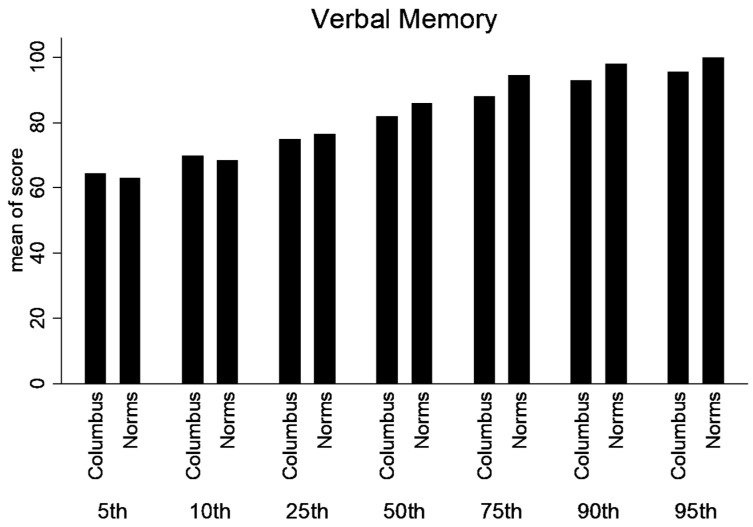
Percentile ranks for Composite Verbal Memory.

**Figure 5 ijerph-21-00247-f005:**
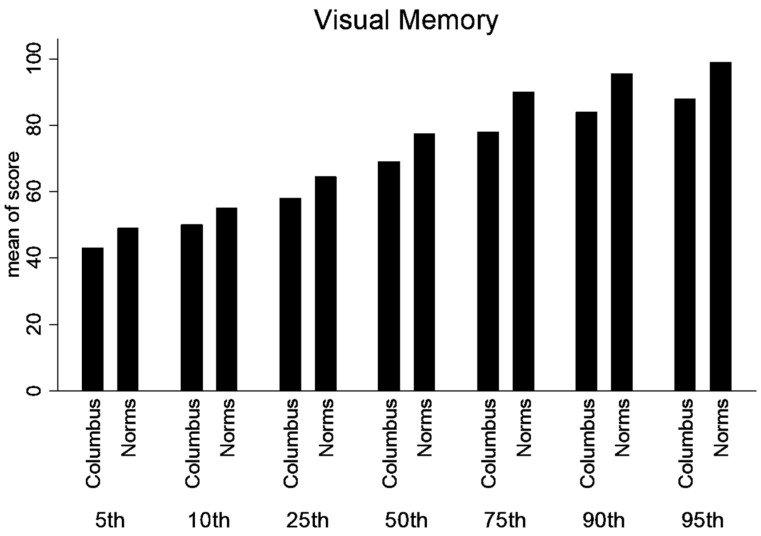
Percentile ranks for Composite Visual Memory.

**Figure 6 ijerph-21-00247-f006:**
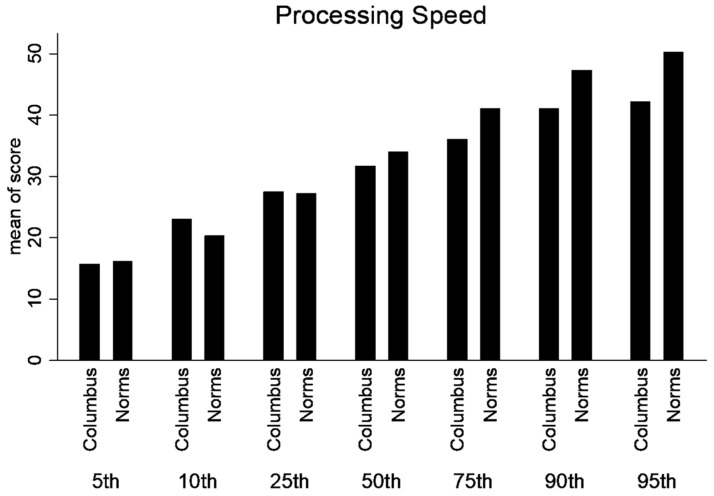
Percentile ranks for Composite Processing Speed.

**Figure 7 ijerph-21-00247-f007:**
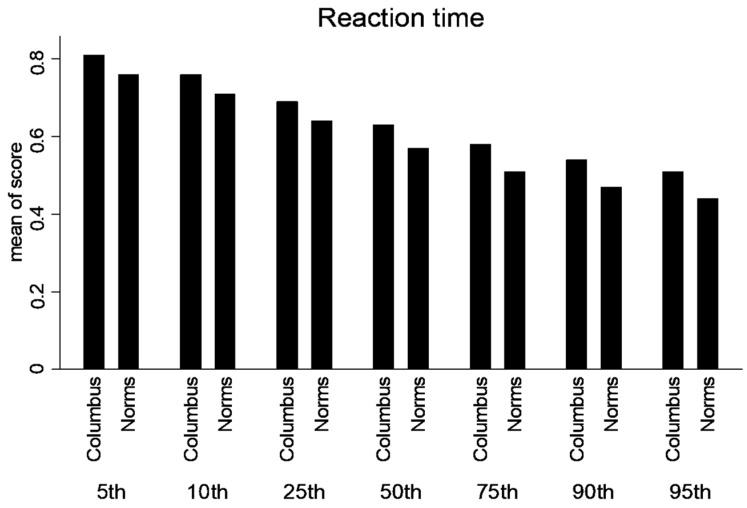
Percentile ranks for Composite Reaction Time.

**Table 1 ijerph-21-00247-t001:** Demographic information of participants.

Age		Height	Weight	Ethnicity	% Free and Reduced Lunch
(Years ± S.D)		(in. ± S.D)	(lbs)	%White	
Total	15.6 ± 2.5	69.3 ± 3.35	180.4 ± 48.31	19%	81%
13–15	14.2 ± 1.1	67.6 ± 3.77	170.6 ± 46.87	20%	81%
16–18	16.9 ± 1.2	70.6 ± 3.43	190.7 ± 50.11	19%	81%

**Table 2 ijerph-21-00247-t002:** Comparison of industry-provided normative data and urban sample on composite scores.

Age	Test	Urban	Published Norm	T	*p*	95% Confident Interval
13–15 years old (*n* = 300)	Verbal	81.3 (9.8)	85	−6.7	<0.001	80.2–82.4
	Visual	67.5 (13.1)	73	−6.9	<0.001	65.9–69.1
	Visual Motor Speed	31.5 (7.4)	38	−15.9	<0.001	30.6–32.3
	Reaction Time	0.65 (0.1)	0.57	11.8	<0.001	0.63–0.66
16–18 years old (*n* = 331)	Verbal	81.9 (9.9)	84	−4.1	<0.001	80.9–82.9
	Visual Motor Speed	34.3 (7.3)	36	−5.7	<0.001	33.4–35.1
	Reaction Time	0.6 (0.09)	0.53	6.9	<0.001	0.59–0.62

## Data Availability

The raw data supporting the conclusions of this article will be made available by the authors on request.

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
