# Peer review of "Comparison of Urban versus Industry Normative Values of Immediate Post-Concussion Assessment and Cognitive Testing (ImPACT™)"

_ijerph, 2024, doi:10.3390/ijerph21030247_

Round 1

Reviewer 1 Report

Comments and Suggestions for Authors

The authors must be commended for carrying out a study regarding the comparison of urban versus industry normative values of ImPACT™. This topic is very interesting and important, the research methodology used in the study is appropriate, and the manuscript is written with good clarity. However, some issues need to be taken into consideration.

Introduction

The introduction part is too long. I strongly suggest shortening it.

Methods

Please explain more briefly the data collection part. What is PPE composed of (which test for physical examination e.g.)? Who conducted the testing? Also, did you conduct the data collection according to previous studies? If so, please emphasize it.

Results

I suggest graphically processing the figures. I think that this kind of paper deserves better figures.

Discussion

Congrats on the discussion section.

Author Response

Please see the attached file that addresses comments from reviewer 1.

Reviewer 2 Report

Comments and Suggestions for Authors

Thank you for your work, I only have a few points of improvement to mention:

- Your paper is important and fills important research gaps, but this is not made sufficiently clear in the introduction for a lay reader.

- The justification for the sample size should be more explicit, addressing statistical power and representativeness.

- Comparing the urban high school athlete study with normative ImPACT™ data is a sound analytical approach to identify potential disparities, but the extent of the differences needs to be analysed to fully capture the magnitude or practical significance of the results.

- El documento mejoraría con un examen crítico de si las diferencias significativas identificadas se traducen en un impacto significativo en el rendimiento de los atletas en el campo o en su recuperación de una lesión.

- The main limitation of this study is the generalisability of the results, as the study bases its findings on a specific urban population. This should be more clearly stated in the text.

- About the conclusions, the authors should aim to reiterate the key findings and offer insight into how this may influence clinical practice or policy. Suggestions for future research should be given context and possibly even prioritize the most pressing questions that emerged from the study.

- As a future research, a longitudinal study design is recommended to track changes over time and establish causation more definitively. 

Author Response

Please see the attached word document for the authors' responses to the comments from reviewer 2. 

Reviewer 3 Report

Comments and Suggestions for Authors

Thank you for submitting manuscript “Comparison of Urban versus industry normative values of ImPACT”. Presenting the research results of the submitted scientific article required many sacrifices and logistics and organization of scientific research. After reviewing mentioned above scientific paper I would like to require some advices.

Below I am sending You outline possible points for revision in the chronological order of the manuscript.

Abstract - please add information in the abstract about the applied value of the research and increase information about the practical application from the research.

Introduction - please add citations to the sentences (line 38-39).                                                        

Introduction - I propose to change the sentence or add them to the discussion – line 101-102 and line 130-132. Authors in mentioned sentence start to discuss about the results of scientific studies, not inform about the all relevant references connected with their scientific paper.  

Table 1. – please change subtitle of the Table 1 – not “Sample Age Height” – it is not scientific language.

Results - there is one thing I would propose to change in the research results section. The authors should put a description of the research results under each graph, this will be clearer for the audience.

Discussion – line 245 – the authors start the sentence from the word “Ip” – should be “If”?

References – please check the following references (lack of number, volume or pages) 6, 21, 31, 58, 59 (add dot.)

Author Response

Please see the attached word document for the authors' responses to the comments from reviewer 3.
